# A Cluster Randomized Controlled Trial of the *Archena Infancia Saludable* Project on 24-h Movement Behaviors and Adherence to the Mediterranean Diet among Schoolchildren: A Pilot Study Protocol

**DOI:** 10.3390/children10040738

**Published:** 2023-04-17

**Authors:** José Francisco López-Gil, Antonio García-Hermoso, Lee Smith, Alejandra Gallego, Desirée Victoria-Montesinos, Yasmin Ezzatvar, Maria S. Hershey, Héctor Gutiérrez-Espinoza, Arthur Eumann Mesas, Estela Jiménez-López, Pedro Antonio Sánchez-Miguel, Alba López-Benavente, Laura Moreno-Galarraga, Sitong Chen, Javier Brazo-Sayavera, Alejandro Fernandez-Montero, Pedro Emilio Alcaraz, Josefa María Panisello Royo, Pedro J. Tárraga-López, Stefanos N. Kales

**Affiliations:** 1Navarrabiomed, Hospital Universitario de Navarra (HUN), Universidad Pública de Navarra (UPNA), IdiSNA, 31006 Pamplona, Spain; 2Department of Environmental Health, Harvard University T.H. Chan School of Public Health, Boston, MA 02138, USA; 3One Health Research Group, Universidad de Las Américas, Quito 170124, Ecuador; 4Centre for Health, Performance and Wellbeing, Anglia Ruskin University, Cambridge CB1 1PT, UK; 5Department of Applied Economics, Faculty of Economics and Business, University of Murcia, 30100 Murcia, Spain; 6Faculty of Pharmacy and Nutrition, UCAM Universidad Católica San Antonio de Murcia, 30107 Murcia, Spain; 7Department of Nursing, Universitat de València, 46007 Valencia, Spain; 8Escuela de Fisioterapia, Universidad de las Américas, Quito 170504, Ecuador; 9Health and Social Research Center, Universidad de Castilla-La Mancha (UCLM), 16071 Cuenca, Spain; 10Postgraduate Program in Public Health, Universidade Estadual de Londrina, Londrina 86057-970, Brazil; 11Grupo Análisis Comportamental de la Actividad Física y el Deporte (ACAFYDE), Departamento de Didáctica de la Expresión Musical, Plástica y Corporal, Facultad de Formación del Profesorado, Universidad de Extremadura, 10003 Cáceres, Spain; 12Departamento de Expresión Plástica, Musical y Dinámica, Facultad de Educación, Universidad de Murcia, 30100 Espinardo, Spain; 13IdiSNA (Instituto de Investigación Sanitaria de Navarra), 31008 Pamplona, Spain; 14Department of Pediatrics, Complejo Hospitalario de Navarra, Servicio Navarro de Salud, 31008 Pamplona, Spain; 15Institute for Health and Sport, Victoria University, Melbourne 8001, Australia; 16PDU EFISAL, Centro Universitario Regional Noreste, Universidad de la República (UdelaR), Rivera 40000, Uruguay; 17Department of Sports and Computer Science, Universidad Pablo de Olavide (UPO), 41013 Seville, Spain; 18Department of Occupational Medicine, University of Navarra, 31008 Pamplona, Spain; 19Research Center for High Performance Sport, San Antonio Catholic University of Murcia, 30830 Murcia, Spain; 20Faculty of Sport Sciences, San Antonio Catholic University of Murcia, 30107 Murcia, Spain; 21Fundación para el Fomento de la Salud, 28006 Madrid, Spain; 22Departamento de Ciencias Médicas, Facultad de Medicina, Universidad Castilla-La Mancha (UCLM), 02008 Albacete, Spain

**Keywords:** physical activity, screen time, sedentary behavior, sleep duration, movement guidelines, movement recommendations, children, lifestyle, parents, family

## Abstract

**Objective:** The aim of this paper is to describe the protocol of pilot cluster randomized controlled trial (RCT) that will evaluate the effects of a lifestyle-based intervention. The *Archena Infancia Saludable* project will have several objectives. The primary objective of this project is to determine the 6-month effects of a lifestyle-based intervention on adherence to 24-h movement behaviors and Mediterranean diet (MedDiet) in schoolchildren. The secondary objective of this project is to test the intervention effects of this lifestyle-based intervention on a relevant set of health-related outcomes (i.e., anthropometric measurements, blood pressure, perceived physical fitness, sleep habits, and academic performance). The tertiary objective is to investigate this intervention’s “halo” effect on parents’/guardians’ 24-h movement behaviors and adherence to the MedDiet. **Methods:** The *Archena Infancia Saludable* trial will be a cluster RCT submitted to the Clinical Trials Registry. The protocol will be developed according to SPIRIT guidelines for RCTs and CONSORT statement extension for cluster RCTs. A total of 153 eligible parents/guardians with schoolchildren aged 6–13 years will be randomized into an intervention group or a control group. This project focuses on two fundamental pillars: 24-h movement behaviors and MedDiet. It will mainly focus on the relationship between parents/guardians and their children. Behavior change strategies for dietary and 24-h movement behaviors in schoolchildren will be based on healthy lifestyle education for parents/guardians through infographics, video recipes, brief video clips, and videos. **Conclusions:** Most of the current knowledge on 24-h movement behaviors and adherence to the MedDiet is based on cross-sectional or longitudinal cohort studies, warranting a need to design and conduct RCTs to obtain more robust evidence on the effect of a healthy lifestyle program to increase 24-h movement behaviors and to improve adherence to the MedDiet in schoolchildren.

## 1. Introduction

The 24-h movement guidelines for youth have shifted the focus from individual physical activity (PA) components to an integration of all movement-related behaviors in the 24-h time-use continuum [1]. These guidelines indicate that children and adolescents (aged 5–17 years) should engage daily in at least 60 min of moderate-to-vigorous PA, restrict their recreational screen time (≤120 min per day for children/adolescents), and obtain adequate sleep duration (e.g., 9–11 h per day for children, 8–10 h for adolescents) in a period of 24 h [1,2]. Thus, clustering and interactions between all 24-h movement guidelines should be promoted to improve health outcomes [3]. PA, sleep duration, and sedentary behavior (including screen time) have been related to a wide range of essential health and developmental outcomes in a young population [4]. To date, most studies have investigated these movement behaviors in isolation. However, recently, attention has turned toward a comprehensive method that acknowledges the interdependence and interconnectedness of 24-h movement behaviors [3,5]. Such guidelines note that the clustering and interactions among all domains of 24-h movement behaviors need to be addressed to enhance health results (e.g., lower risk of obesity, type 2 diabetes, depression and suicidal ideation, higher physical fitness) [3,6,7,8,9]. Despite these attributed health benefits, a recent meta-analysis reported a global rate of adherence to the 24-h movement guidelines of only 7.12% [10].

Concerning eating habits, the World Health Organization (WHO) advises a healthy diet to help protect against malnutrition as well as noncommunicable diseases such as cancer, cardiovascular diseases, stroke, and type 2 diabetes [11]. However, the need to improve dietary quality on a global scale has recently been highlighted [12]. In this sense, the Mediterranean Diet (MedDiet) is a dietary pattern well known worldwide for its distinctive characteristics and health benefits [13,14,15]. The MedDiet includes an eating pattern rich in fruits and vegetables (seasonal), legumes, nuts, whole grains, and olive oil as the main dietary fat with a greater intake of white or lean meats rather than red or processed meats, moderate consumption of dairy products (cheese, milk), moderate consumption of fish, eggs, and small intakes of wine with meals (only in adults) [16,17]. Supporting this notion, scientific evidence has demonstrated the inverse relationship between the MedDiet and noncommunicable diseases in adults (e.g., cancer, metabolic syndrome, hypertension, and cardiovascular diseases) [18]. Specifically, among young people, greater adherence to the MedDiet has been associated with greater anti-inflammatory potential [19]. Unfortunately, even though it is a healthy dietary pattern supported by evidence, a systematic review has pointed out the clear trend of decreasing adherence to the MedDiet in Mediterranean countries, especially among children [20,21]. For instance, Cabrera et al. [20] reported an overall rate of only 10% with high adherence to the MedDiet among a young population. As possible reasons for this trend, westernization of diets [22], urbanization [23], lifestyle changes [24,25], economic factors [26], and lack of knowledge and education [27] (among others) have been proposed.

Previous studies have found an association between meeting the 24-h movement guidelines and healthier dietary patterns [28,29]. Regarding MedDiet, only one previous cross-sectional study (in adolescents) has found that meeting all three 24-h movement guidelines has been related to greater adherence to the MedDiet compared to those who did not meet these guidelines [30]. It is possible that the low prevalence of 24-h movement guidelines reported worldwide [10] and in Spain [28] may be another factor to be taken into account in the low adherence to the MedDiet currently described [20,21]. Although the need for interventions focusing on improving adherence to the 24-h movement guidelines [10] and MedDiet [25] has been previously suggested, the literature on this matter is limited. More specifically, to our knowledge, no previous randomized controlled trial (RCT) has verified the effect of a healthy lifestyle program on both adherence to 24-h movement behaviors and the MedDiet in schoolchildren. In addition, although associations between 24-h movement behaviors or adherence to the MedDiet and numerous health-related outcomes (e.g., blood pressure, obesity-related indicators, physical fitness, sleep quality) have been previously reported, there is a lack of scientific evidence on the effect of healthy lifestyle interventions, including their interaction effects on particular health-related outcomes in addition to academic performance, when both are implemented at the same time. This also denotes the need for well-designed RCTs focused on this matter.

The objective of this paper is to describe the protocol of a pilot cluster RCT that will evaluate the effects of a lifestyle-based intervention. The *Archena Infancia Saludable* project will have several objectives. The primary objective of this project is to determine the 6-month effects of a lifestyle-based intervention on 24-h movement behaviors and adherence to the MedDiet in schoolchildren. Furthermore, the secondary objective of this project is to test the intervention effects of this lifestyle-based intervention on a relevant set of health-related outcomes (i.e., anthropometric indicators, resting blood pressure and heart rate, sleep problems, health-related quality of life, perceived physical fitness, and academic performance). Likewise, the tertiary objective of this study is to verify this intervention’s “halo” effect on 24-h movement behaviors and adherence to the MedDiet. We hypothesize that the *Archena Infancia Saludable* project will achieve improvements with small-to-medium effects on 24-h movement behaviors and adherence to the MedDiet in schoolchildren.

## 2. Materials and Methods

### 2.1. Design

The *Archena Infancia Saludable* (Figure 1) project will be a cluster randomized, parallel-group, clinical trial. The protocol was developed according to SPIRIT guidelines for RCTs [31] and CONSORT statement extension for cluster RCTs [32].

### 2.2. Setting

#### 2.2.1. Procedure

The unit of randomization, intervention, and cluster analysis are the participating parents/guardians with schoolchildren aged 6–13 years, who will be randomized into an intervention group or a control group. The study will be conducted at four different times during one academic year:(a)*First phase.* For four months, we will prepare protocols, set up measurement techniques, enroll the study participants, and collect the baseline data from both parents/guardians and their children.(b)*Second phase*. The intervention program will be conducted for six months.(c)*Third phase*. For one month, we will collect postintervention data from both parents/guardians and their children.(d)*Fourth phase*. In the last month, the control group will receive all the contents of the healthy lifestyle program upon completion of the program by the intervention group.

#### 2.2.2. Rationale for the Age Group Chosen

This project will target schoolchildren aged 6–13 years. This age group was chosen because childhood is a critical period for adopting daily routines and habits. In addition, the *Archena Infancia Saludable* program will be focused on parents/guardians because they are in a key position to encourage healthy behaviors among their children [33]. Furthermore, an additional reason that justifies this choice of study population lies in the low adherence to the MedDiet [20,21,25] and meeting all the 24-h movement guidelines [4,10] reported in schoolchildren.

#### 2.2.3. Schoolchildren Eligibility

Regarding the inclusion criteria, schoolchildren aged 6−13 years will be eligible. The exclusion criteria will be defined as follows: (a) participants with any pathology that contraindicates exercise or that requests special attention; (b) participants under pharmacological treatment that prevents them from receiving the contents of the activities of the program; (c) participants or parents/guardians presenting Spanish learning difficulties in understanding the contents of the questionnaires; (d) participants whose parents/guardians do not authorize participation in the research project; or (e) participants who decline to participate in the research project.

#### 2.2.4. Recruitment and Randomization

In this pilot study, recruitment will be performed in one school randomly selected from Archena (Region of Murcia, Spain). Previously, we will contact the directors of all schools of Archena (Region of Murcia, Spain), and we will release announcements through local media channels. Any parent/guardian with a child who meets the inclusion criteria indicated above will be invited to participate. A blinded randomization of the participants into the intervention or control group will be performed using the list of encrypted codes of the participants using the software SPSS (IBM Corp, Armonk, NY, USA) (version 25.0) for Windows. To decrease the risk of selection bias during the assessments, a researcher who will not participate in either the data collection or in the statistical analysis will be responsible for randomizing the groups after the intervention. This process will be conducted immediately after the collection of baseline data. The researchers who will participate in the data collection will not know to which group the schoolchildren belong, neither at the baseline nor at the postintervention measurements.

### 2.3. Intervention

The intervention group will complete the *Archena Infancia Saludable* interdisciplinary program for six months. The interdisciplinary research team comprised nutritionists, physicians, PA and sports science professionals, physical education teachers, nurses, and psychologists. The investigators responsible for carrying out the intervention program will not participate in data collection or statistical analysis, so they will not be aware of the participants’ group assignment.

The *Archena Infancia Saludable* project will focus on two fundamental pillars: 24-h movement behaviors and MedDiet. Some examples of the contents of the intervention program are shown in Figure 2. The project will mainly focus on the relationship between parents/guardians and their children. Furthermore, the program includes a behavioral approach (i.e., nutritional education), which encourages the responsibility among all the participants to maintain healthy behavioral changes over the long term [19,34].

The nutritional approach will follow the MedDiet model [13]. We will not impose any caloric restriction since we aim to establish a healthy diet based on the MedDiet [14,25]. Concerning the 24-h movement behaviors, parents/guardians will be told to encourage their children to adopt an active lifestyle with a daily equilibrium of PA, sedentary behavior, and sleep that will support their healthy development [1]. Dietary and 24-h movement behavior changes in schoolchildren will be based on healthy lifestyle education for parents/guardians by infographics, video recipes, information pills, and videos. The contents of these materials have been created by the research team following international and national guidelines for PA [1,35,36], sedentary behavior [1], sleep duration [1,37], MedDiet [38], and healthy eating guidelines [39,40].

The intervention will be performed by the communication application for schools TokApp (TokApp Online S.L., Vigo, Spain). All the contents of the intervention programs will be delivered online. In line with our hypothesis, only parents/guardians will receive the contents of the intervention. We will try to verify whether intervening in parents/guardians has an impact on 24-h movement behaviors and adherence to the MedDiet in their children. Thus, parents/guardians will receive three different contents weekly (i.e., infographics, video recipes, information pills, or videos) related to 24-h movement behaviors (i.e., PA, sedentary behavior, sleep duration) or MedDiet. In addition, each launched content will be available until the end of the intervention for those who have not been able to view it at the time of its launch. To not discriminate participants from the control group, all project materials will be offered to parents/guardians allocated to the control group at the end of the intervention phase as well. The researchers responsible for sending the contents of the intervention program through the communication application TokApp will not participate in data collection or statistical analysis after the intervention.

The length of the intervention (24 weeks during a school-academic year) falls within the time range used in the prior RCTs conducted on this topic (i.e., 20 weeks [41] and 60 weeks [42]). It has been previously described that when an intervention program is delivered, it can have a compensatory effect, so that participants discontinue other physical activities that they would normally have done otherwise. To address this issue, we will assess PA using activity monitors (i.e., accelerometers) for seven days at two different times during the study: at baseline and postintervention. Finally, any adverse effects will be documented and reported with trial outcomes.

### 2.4. Strategies to Enhance Adherence to the Intervention Program

Parents/guardians will receive a verbal invitation to take part in the study intervention program and to refer to all the assessments and contents. They will receive a reminder if they have not viewed the contents sent after one week. Our goal will be that parents/guardians engage in at least 80% of the weekly content, which will be considered a successful attendance rate. This evaluation will be possible because the communication application TokApp allows us to know the interaction with the content sent. However, we will encourage the schoolchildren and their families to visualize all contents weekly whenever possible.

### 2.5. Statistical Procedures

#### 2.5.1. Sample Size

The sample size calculation was performed following the indications by Donner et al. [43]. First, we calculated the sample sizes without adjustment for clustering (*N_0_*). For this purpose, the statistical analysis in this study involved several parameters, including the threshold probability for rejecting the null hypothesis (α), which represents the type I error rate, and the probability of failing to reject the null hypothesis under the alternative hypothesis (*β*), representing the type II error rate. In addition, the proportion of subjects in the intervention group (*q*_1_) and control group (*q*_0_), the effect size (Cohen’s *d*), and the standard deviation of the outcome in the population (*σ*) were calculated. Thus, an *α* value (two-tailed) of 0.05 and a *β* value of 0.20 were established so that the standard normal deviation for *α* is *Z_α_* = 1.960 and for *β* is *Z_β_* = 0.842. The proportion of subjects in both the intervention and control groups will be similar (*q*_1_ = 0.50; *q*_0_ = 0.50). Our study will be powered to detect medium-sized effects (i.e., Cohen’s *d* = 0.5), and the standard deviation of the outcome in the population will be 1.0.
$$N_{0}=\frac{\left(\frac{1}{q_{1}}+\frac{1}{q_{0}}\right)\;*\;(\left(Z_{\alpha \;}+Z_{\beta }{)}^{2}\right)}{{\left(\frac{d}{\sigma }\right)}^{2}}$$
$$N_{0}=\frac{\left(\frac{1}{0.50}+\frac{1}{0.50}\right)*(\left(1.960+0.842{)}^{2}\right)}{{\left(\frac{0.5}{1.0}\right)}^{2}}$$
*N*_0_ = 125.58 ≈ 126 participants

Second, we calculated the sample size with adjustment for clustering (*N*_1_). For previous studies in this specific population, we assumed an average household size (*m*) of 1.2 participants according to previous studies performed in the same region [44,45]. Moreover, the within-cluster correlation coefficient (*ρ*) was established as 0.5.
Design effect = 1 + (*ρ* × (*m* − 1))
Design effect = 1 + (0.5 × (1.2 − 1)) = 1.1
*N*_1_ = *N*_0_ × Design effect => 138.14 ∗ 1.1 = 138.142 ≈ 138 participants

Third, we further assumed a 10% drop-out rate, which was estimated using the following formula:*N*_3_ = *N*_2_ / (1 − % losses)
*N*_3_ = 138/(1 − 0.1) = 153 participants

Thus, a minimum sample size of 153 participants for the whole sample will be required to perform this study.

#### 2.5.2. Statistical Analysis

The mean (*M*) and standard deviation (*SD*) will be provided for quantitative variables, while frequencies (*n*) and percentages (%) will be given for qualitative variables. Kolmogorov-Smirnov’s test with Lilliefors correction and Levene’s test will be used to assess the normality of data and the equality of variances, respectively. Thereafter, Student’s *t* test or Mann–Whitney’s *U* test for two-group comparisons will be used in relation to meeting the normality assumption. Pearson’s chi-square (χ^2^) test will be employed to verify associations between qualitative variables. For quantitative variables, the relationship will be tested through Spearman’s rho (*ρ*) or Pearson’s *r*, based on the normality assumption. Preliminary analyses will be conducted to identify the frequency, range, variability, and distribution pattern of each variable to apply the most appropriate statistical test when comparisons are needed. Since this RCT has an experimental design with two data collections of the primary, secondary, and tertiary outcomes, the first at baseline (*t*_0_ = 0 weeks) and the second after intervention (*t*_1_ = 24 weeks) in both the intervention and control groups, we will apply a comparative analysis between these measures to establish differences between groups. To evaluate the intervention effect, multilevel mixed-effects regression models with repeated measures will be conducted for each dependent variable. Then, multivariate analyses will be performed, considering the autocorrelation between repeated measures. Both intention-to-treat (ITT) (which measures the effect of assigning an intervention) and per-protocol (PP) analysis (which measures the effect of receiving an intervention) approaches will be applied for the data analysis. All statistical analyses will be conducted using Stata software (Stata, College Station, TX, USA) (version 17.0) for Windows. A *p* value ≤ 0.05 will determine statistical significance.

### 2.6. Variables

The complete set of primary, secondary, and tertiary outcomes will be assessed twice, at the time of enrollment and after the 24-week healthy lifestyle program. The measurements will be performed at school by evaluators previously trained to standardize the measurements and blinded to the group in which participants will be allocated. A summary of all the variables that will be examined in the *Archena Infancia Saludable* project is provided in Table 1.

#### 2.6.1. Primary Outcomes (Schoolchildren)

##### 24-h Movement Behaviors (Accelerometers)

PA and sedentary time will be evaluated by accelerometers. A triaxial accelerometer (Actigraph GT3x, Pensacola, FL, USA) will be used to assess PA, sedentary time, and sleep duration over seven consecutive days. Participants will be instructed to wear the device attached to the nondominant wrist. Since there are nonwaterproof devices, schoolchildren will wear the accelerometers 24 h a day and will only be able to remove them while bathing or swimming. In addition, schoolchildren will have a paper-based diary log to record the time when they go to bed, wake up, and remove the device.

##### 24-h Movement Behaviors (Self-Reported)

Physical activity will be measured using a self-report questionnaire called the Youth Activity Profile—Spain (YAP-S) [46] aimed at assessing physical activity levels and sedentary behaviors in children. The questionnaire was designed to be completed by the children themselves and involves recalling activities from the past seven days. Screen time will be measured in children by asking their parents/guardians about the amount of time the children spend engaged in sedentary screen-based activities as follows: “Approximately, how much time does your child typically spend in front of a screen (on daily average), including computer, tablet, television, videos, video games or cell phone screen?”. The parents or guardians will be asked separately about screen time on weekdays and weekends. Specifically, they will be asked about the average daily amount of time their child spends in front of a screen. To calculate an overall screen time score, the responses to the three questions (two for the weekend and one for weekdays) will be added together after weighting. Sleep duration will be measured by asking parents/guardians about their child’s bedtime and wake-up time separately for weekdays and weekends as follows: “What time does your child usually go to bed?” and “What time does your child usually get up?”. The average daily sleep duration will be computed for each participant by adding together the average nocturnal sleep duration on weekdays (multiplied by 5) and the average nocturnal sleep duration on weekends (multiplied by 2) and then dividing the total by 7. In addition, two additional questions will be asked about *siesta* habits. First, parents/guardians will be asked if their child usually takes a *siesta*, with options of yes or no. Second, parents/guardians will be asked about the duration of the child’s *siesta*, with options ranging from 0–15 min to 120 min or more. A *siesta* is a customary short nap that is usually taken in the early afternoon, typically following the midday meal, and is prevalent in countries with hot weather conditions, such as those found in the Mediterranean area. The school schedule for all schoolchildren is from 09:00 a.m. to 14:00 p.m., which allows time for children to take a *siesta* after this period.

##### Adherence to the Mediterranean Diet

To assess adherence to the MedDiet, the Mediterranean Diet Quality Index for Children and Teenagers (KIDMED) will be used, which has been previously validated in a young Spanish population [14]. The KIDMED ranges from −4 to 12 and is based on a 16-question test. Items reporting unhealthy characteristics related to the MedDiet are scored with −1 point, and those reporting healthy characteristics with +1 point. The sum of all scores from the KIDMED test will be used to categorize into three different levels of adherence: (a) optimal MedDiet (≥ 8 points), (b) improvement needed to adjust intake to Mediterranean patterns (4−7 points), and (c) very low diet quality (≤ 3 points) [14].

#### 2.6.2. Secondary Outcomes (Schoolchildren)

##### Anthropometric Measurements

The weight of the schoolchildren will be assessed using a Tanita BC-545 electronic scale with an accuracy of 0.1 kg, while height will be measured with a Leicester Tanita HR 001 portable height rod with a precision of 0.1 cm. Body mass index (BMI) will be computed by dividing body weight in kilograms by height in squared meters. Additionally, BMI z scores will be determined using the sex- and age-specific thresholds provided by the World Health Organization [47] and the International Obesity Task Force Criteria [48]. The BMI z scores will then be utilized to identify excess weight (i.e., overweight or obesity). Furthermore, waist circumference will be measured to the nearest 0.1 cm at the umbilical level using a measuring tape with a standardized level of tension applied throughout the measurement. The waist-to-height ratio (WHtR) will be calculated, and a WHtR value ≥0.5 will be considered an indicator of abdominal obesity [49].

##### Active Transportation

Active transportation to and from school will be determined by a self-report questionnaire of the PACO (*Pedalea y Anda al Cole*) project [50]. Participants will respond to the following questions: “How do you usually go to school?”, and “How do you usually come back from school?”. Additionally, the second group of inquiries pertains to the method of transportation used for traveling to and from school on a weekly basis. The possible responses include walking, by car, bike, motorbike, bus, or other transport (requesting specific open-ended information in this case).

##### Resting Blood Pressure and Heart Rate

For the purpose of measuring resting blood pressure and heart rate in schoolchildren, an automated blood pressure monitor will be used, which includes a cuff of appropriate size (Omrom^®^ EVOLV HEM-7600T-E, Health-care Co, Kyoto, Japan). The children will be asked to sit in a quiet room with their feet on the ground and back supported for 10 min, after which two readings of their blood pressure will be taken, with the second reading being taken five minutes after the first. The average of the two readings for systolic and diastolic blood pressure will be preserved, and mean arterial pressure will be calculated using the following formula: diastolic blood pressure + [0.333 × (systolic blood pressure—diastolic blood pressure)]. The blood pressure measurements will be categorized using age-, sex-, and height-specific cutoff points provided by the European Society of Hypertension guidelines for children and adolescents [51]. High-normal blood pressure will be defined as systolic or diastolic blood pressure that is equal to or greater than the 90th percentile but less than the 95th percentile for young people aged 0–15 years. Hypertension or percentile hypertension will be defined as systolic or diastolic blood pressure that is equal to or greater than the 95th percentile for young people aged 0–15 years.

##### Sleep-Related Problems

The BEARS (B = Bedtime Issues, E = Excessive Daytime Sleepiness, A = Night Awakenings, R = Regularity and Duration of Sleep, S = Snoring) scale will be used to assess sleep-related issues. The BEARS scale is a screening tool that includes questions related to bedtime problems, excessive daytime sleepiness, night awakenings, regularity and duration of sleep, and snoring. This tool will be used in an interview to screen for the most common sleep-related problems in children and adolescents [52]. The parents/guardians will report the results of the BEARS scale. The Spanish version of the BEARS scale has been found to be valid for screening sleep-related problems in pediatric evaluations [53].

##### Health-Related Quality of Life

Health-related quality of life will be measured by the Child Health Utility 9D (CHU9D) [54,55]. The questionnaire was originally intended for children aged 7–11 years, but it can also be applied for children who are 6 years old with the help of an interviewer [56]. The CHU9D comprises 9 different areas, including worry, sadness, pain, tiredness, annoyance, school-work/homework, sleep, daily routine, and the ability to participate in activities. Each area has 5 different levels indicating increasing severity. The scores from the CHU9D will be utilized in cost-utility analyses [57]. In addition, cost-effectiveness will be assessed by an ad hoc questionnaire answered by parents/guardians, including the number of days in the hospital, pediatrician visits, medicine use and its cost, and study days lost due to health problems during the last 24 weeks.

##### Self-Reported Physical Fitness

The International Fitness Scale (IFIS) will be used to evaluate self-reported physical fitness. This scale comprises five items that use a 5-point Likert scale to ask about the children’s overall perceived physical fitness, as well as their perceived cardiorespiratory fitness, muscular fitness, speed-agility, and flexibility when compared to their peers. The Likert scale includes options for very poor, poor, average, good, and very good physical fitness [58].

##### Academic Performance

At the end of the academic year, the school will provide academic records. Academic performance will be evaluated by two different methods. First, it will be assessed based on the grades obtained in Mathematics, Language, Mathematics and Language combined, and English, as well as the grade point average of these three subjects. These measurements have been used in previous studies as indicators of academic performance [59,60]. Second, academic performance will be evaluated by calculating the average of all the subjects taken by the schoolchildren [61].

#### 2.6.3. Tertiary Outcomes (Parents/Guardians)

##### 24-h Movement Behaviors (Self-Reported)

The level of PA and sitting time will be assessed by the International Physical Activity Questionnaire-short form (IPAQ-SF) [62]. The IPAQ-SF will gather information on physical activity (PA) during the previous week in terms of the number of days, duration, and intensity of vigorous-intensity, moderate-intensity, and walking activities that lasted for at least 10 min. It will also inquire about the amount of time spent sitting on the last seven weekdays. The IPAQ-SF score will be computed in Metabolic Equivalent of Task (MET)-minutes per day or week. Parents/guardians will be classified into meeting or not meeting the PA guidelines. Meeting the PA guidelines will be considered when parents/guardians engage in ≥150 moderate-vigorous PA minutes per week or ≥75 vigorous-intensity PA minutes per week. For the sitting time, participants will be questioned about their sitting time separately for weekdays and weekend days. The 24-h movement guidelines offer an explicit guideline about recreational screen time: ≤3 h daily as a subcomponent of the sedentary guideline (i.e., in addition to the less than 8-h sedentary time guideline) [63]. In this project, recreational screen time will be self-reported. Participants will be asked a series of questions during a household interview regarding their sedentary activities during leisure time as follows: (a) “In a typical week in the past three months, how much time did you usually spend on a computer, tablet or iPad including watching videos, playing computer games, emailing or using the Internet?”; (b) “In a typical week in the past three months, how much time did you usually spend playing other types of video games on a game console or hand-held electronic device?”; (c) “In a typical week in the past three months, how much time did you usually spend watching television, DVDs or videos?” Respondents will be provided with a continuous response option. The amount of time spent on each activity involving screens will be added together to obtain an overall estimate of daily recreational screen time. Screen time will be categorized into meeting (vs. not meeting) the guideline. Sleep duration will be self-reported by using the following question for weekdays and weekends independently: “How many hours do you usually spend sleeping in a 24-h period, excluding time spent resting?” Responses will be reported as a continuous variable and rounded to the nearest half-hour by the interviewer. Sleep duration will be categorized as a binary variable to compare parents/guardians meeting age-specific sleep guidelines (i.e., 7–9 h daily for adults aged 18–64 years; 7–8 h daily for adults aged ≥65 years) with those not meeting that guideline [64].

##### Adherence to the Mediterranean Diet

To assess adherence to the MedDiet, parents/guardians will also be administered the 17-item energy-restricted Mediterranean Adherence Screener (er-MEDAS) [64], used in the PREDIMED (*PREvención con DIeta MEDiterránea*)-Plus trial [65]. The er-MEDAS was developed using the 14-item MEDAS [66], which had been previously validated. Each of the 17 items that pertain to typical dietary habits will be evaluated and assigned a score of either zero or one point. Therefore, the global score will range from zero to 17 points, with zero points denoting no adherence to the MedDiet and 17 indicating maximum adherence to the MedDiet [65].

#### 2.6.4. Covariates

Age and sex will be self-reported by schoolchildren and parents/guardians, respectively. Parents/guardians will be asked for country birth (of their children and themselves) and their marital status. The somatic maturity of schoolchildren will be estimated according to the prediction models proposed by Moore et al. [67]. The socioeconomic status (SES) of the participants will be evaluated using the Family Affluence Scale (FAS-III) [68], which will be answered by the parents or guardians. The FAS-III comprises six items related to household possessions such as vehicles, computers, bathrooms, dishwashers, and family travel. The score will be calculated by summing up the responses, ranging from 0 to 13 points. The final score will determine the participants’ SES category as low (0–2 points), medium (3–5 points), or high (≥6 points). Information about the educational level of parents/guardians will be requested. Possible options will be (a) incomplete primary education, (b) complete primary education, (c) incomplete secondary education, (d) complete secondary education, (e) incomplete higher education, or (f) complete higher education. The body weight and body height of the parents/guardians will be self-reported. Body mass index (BMI) will be computed by dividing body weight (in kg) by height (in squared meters). Subsequently, BMI status will be established by the WHO criteria [69] as follows: underweight <18.5, normal weight 18.5–24.99, overweight 25–29.99, or obesity ≥30. Parents’/guardians’ perception of their child’s body mass index status will be evaluated with the following question: “In relation to his/her height, which of the following options best describes your child’s body weight?: (1) substantially above normal, (2) slightly above normal, (3) normal, (4) below normal” [70]. This same question will be done four their perception of their own body mass index status as follows: “According to your height, which of the next options best describes your body weight: (1) substantially above normal, (2) slightly above normal, (3) normal, (4) below normal?” [70].

### 2.7. Ethical Considerations and Dissemination

The *Archena Infancia Saludable* project has been previously registered in (ClinicalTrials.gov ID NCT05620303) and has been approved by the Ethics Committee of the Albacete University Hospital Complex and the Albacete Integrated Care Management (ID 2202–132). Similarly, this trial will be performed in accordance with the Helsinki Declaration and respecting the human rights of the participants involved.

All study participants will receive written informed consent. All participants will be informed that they have the right to withdraw from the study at any point without giving reason. The results of this project will be disseminated to academic audiences by presentations at national and international conferences and through peer-reviewed publications in relevant journals. The results will be disseminated to the general population, academic audiences, and policy makers and through seminars, social networks, and press releases.

## 3. Discussion

The *Archena Infancia Saludable* project will verify, for the first time, whether a cluster RCT based on 24-h behaviors and adherence to the MedDiet aimed at parents produces improvements in these healthy behaviors among their children. In childhood, unhealthy lifestyle behaviors (e.g., physical inactivity, excessive sedentary time, short sleep duration, unhealthy diet) share several factors in common [71]. They are (1) cumulative; (2) associated with poorer health in adulthood and increased risk of chronic diseases; (3) preventable and a consequence of not carrying out daily health-promoting activities; and (4) influenced by parenting confidence and skills. Despite this, it has been found that most parents/guardians have positive intentions to support their children’s health behaviors, and yet many are unable to promulgate this support [72]. Therefore, parents need innovative and attractive strategies that are not time-consuming and are adapted to the stressful pace of life today.

Implementing healthy lifestyle habits during childhood is a crucial opportunity for primordial prevention [73,74]. In regard to behavioral interventions, schools provide an ideal setting for effective knowledge transfer and education [75,76]. Additionally, involving families in the intervention can create a supportive environment at home and potentially enhance the impact of the intervention [77]. In general, interventions have tended to focus on health education and the provision of guidelines or recommendations with limited or no training of parents and little recognition of the importance of the role of parents in developing healthy lifestyle habits [71]. In this sense, the *Archena Infancia Saludable* program will seek to provide parents/guardians with numerous practical resources based on 24-h movement behaviors and MedDiet that can be applicable and feasible for their children on a daily basis and adjusted to their real context, beyond theoretical concepts and contents.

Guidelines or recommendations from institutions and scientific experts strongly encourage a MedDiet as a healthy eating pattern that could diminish the risk of chronic noncommunicable diseases since childhood [38,39]. This fact is also consistent with the recommendation of meeting with 24-h movement guidelines [1,2], since the current evidence suggests that these guidelines may have essential implications for health and are linked with several desirable health outcomes in young [65] and adult populations [6,7]. Based on the low levels of the young population meeting the 24-h movement guidelines as well as low adherence to the MedDiet (especially in this region of Spain [45]), it is clear that knowledge, awareness, and implementation of these healthy behaviors by the general population need to be improved in combination with one another for their potential synergistic effect [4,10,28,78].

On the other hand, the United Nations and the WHO have established nine international objectives for noncommunicable diseases to achieve by 2025 [79]. Correspondingly, the target of Sustainable Development Goal 3.4 is to decrease the number of deaths caused by noncommunicable diseases by one-third and support well-being and mental health by 2030 [80]. Unhealthy eating habits and insufficient physical activity are among the primary factors causing disability and a considerable number of chronic noncommunicable diseases worldwide [12,81,82]. Providing intervention programs based on scientific evidence (such as the *Archena Infancia Saludable* program) to improve dietary patterns and lifestyles seems necessary, as it could play an important role in public health.

Additionally, it is worth mentioning that in this cluster RCT of the *Archena Infancia Saludable* project, only one school will be selected. This decision is based mainly on the intention to carry out this intervention program as a pilot test to obtain a more consolidated and effective version of the *Archena Infancia Saludable* program. For this purpose, we will try to obtain feedback from parents/guardians and teachers on the functioning of the project, such as suggestions for improvement and barriers to adherence to the intervention program. We will also try to seek additional sources of funding to increase the resources and methodological quality of this cluster RCT, including all the knowledge acquired in this pilot study, all other schools in the municipality, as well as a larger sample of participants. The ultimate goal of this project is to consolidate the intervention program to be implemented on a scaled basis in all schools in the municipality in future editions, as is being done in projects carried out in secondary schools in Archena that include a larger sample [44].

## 4. Conclusions

Most of the current knowledge on 24-h movement behaviors and adherence to the MedDiet has been based on cross-sectional or longitudinal cohort studies, which warrants further evidence by conducting well-designed RCTs to assess the effect of a healthy lifestyle program on adherence to 24-h movement behaviors and the MedDiet in schoolchildren.

## Figures and Tables

**Figure 1 children-10-00738-f001:**
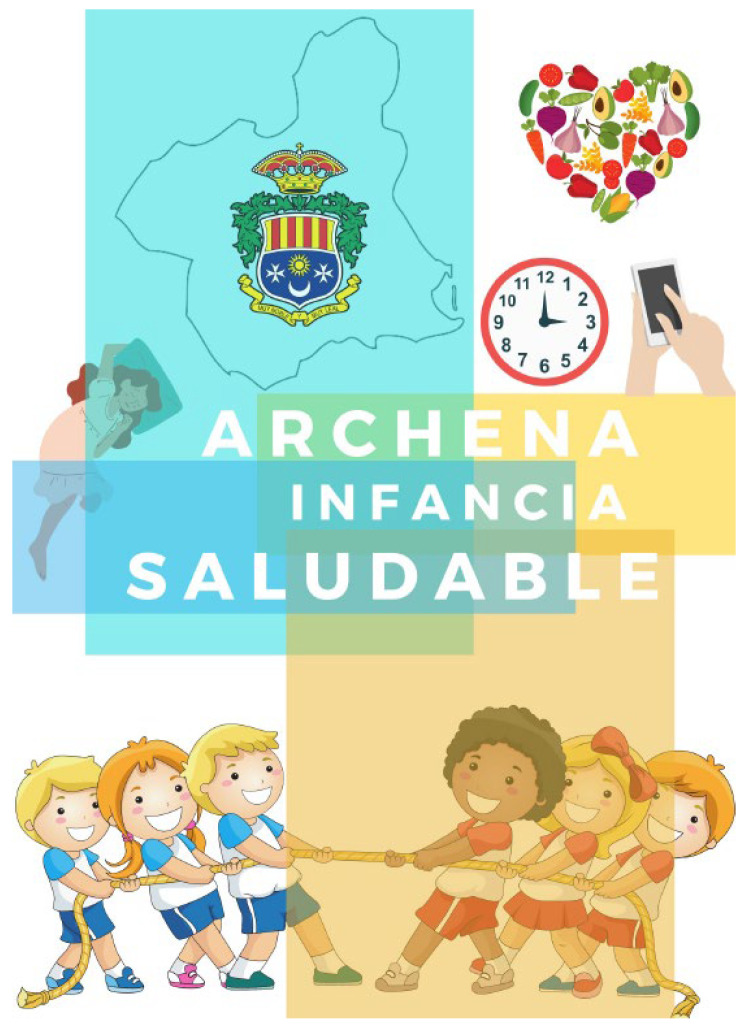
The *Archena Infancia Saludable* project.

**Figure 2 children-10-00738-f002:**
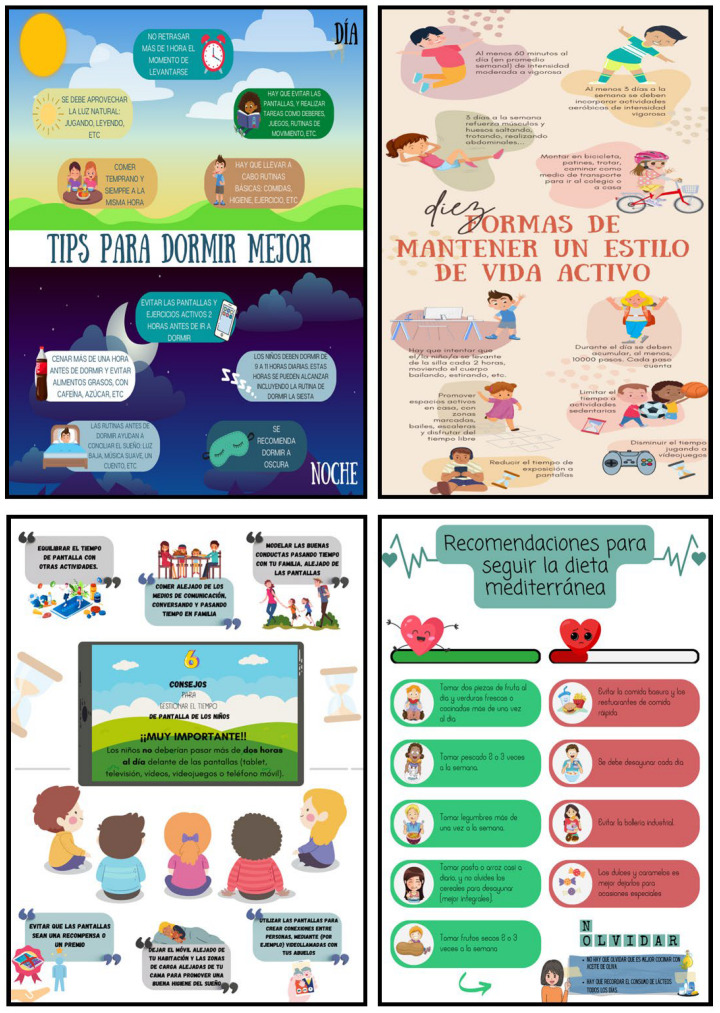
Examples of infographics used in the *Archena Infancia Saludable* project.

**Table 1 children-10-00738-t001:** Summary of the variables examined in the *Archena Infancia Saludable* project.

Variables	Schoolchildren Variables		Parents or Guardians Variables	
	Measurement	Tool	Measurement	Tool
Age	- Self-reported.	- Ad hoc questionnaire.	- Self-reported.	- Ad hoc questionnaire.
Sex	- Self-reported.	- Ad hoc questionnaire.	- Self-reported.	- Ad hoc questionnaire.
Socioeconomic status	- Objective.	- FAS-III.		
Educational level			- Self-reported.	- Ad hoc questionnaire.
Immigrant status	- Objective.	- Ad hoc questionnaire.		
Marital status			- Self-reported.	- Ad hoc questionnaire.
Perception of their child’s BMI status			- Self-reported.	- Ad hoc questionnaire.
Perception of their own BMI status			- Self-reported.	- Ad hoc questionnaire.
Active transportation	- Self-reported.	- PACO questionnaire.		
Resting blood pressure and heart rate	- Objective.	- Omrom^®^ EVOLV HEM-7600T-E.		
Sleep-related problems	- Proxy-reported.	- BEARS scale.		
Health-related quality of life	- Self-reported.	- CHU9D.		
Self-reported physical fitness	- Self-reported.	- IFIS scale.		
Academic performance	- Objective.	- School records.		
Anthropometric measures	- Objective.	- Tanita BC-545, Leicester Tanita HR 001. - Constant tension tape.	- Self-reported.	-Ad hoc questionnaire.
Physical activity	- Objective. - Self-reported.	- Actigraph GT3x. - YAP-S.	- Self-reported.	- IPAQ-short form.
Sedentary behaviors	- Objective. - Self-reported.	- Actigraph GT3x. - YAP-S.	- Self-reported.	- IPAQ-short form.
Recreational screen time	- Proxy-reported.	- Ad hoc questionnaire.	- Self-reported.	- IPAQ-short form.
Sleep duration	- Objective. - Proxy-reported	- Actigraph GT3x. - Ad hoc questionnaire.	- Self-reported.	- Ad hoc questionnaire.
*Siesta*	- Objective.	- Ad hoc questionnaire.		
Adherence to the MedDiet	- Self-reported.	- KIDMED.	- Self-reported.	- PREDIMED questionnaire.

BEARS, Bedtime Issues, Excessive Daytime Sleepiness, Night Awakenings, Regularity and Duration of Sleep, Snoring; BMI, body mass index; CHU9D, Child Health Utility 9D; FAS-III, Family Affluence Scale-III; IFIS, International Fitness Scale; IPAQ, International Physical Activity Questionnaire; KIDMED, Mediterranean Diet Quality Index for Children and Teenagers; MedDiet, Mediterranean diet; PACO, *Pedalea y Anda al Cole*; PREDIMED, PREvención con DIeta MEDiterránea; YAP-S; Youth Activity Profile—Spain.

## Data Availability

Not applicable.

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
