# Peer review of "A Cluster Randomized Controlled Trial of the Archena Infancia Saludable Project on 24-h Movement Behaviors and Adherence to the Mediterranean Diet among Schoolchildren: A Pilot Study Protocol"

_children, 2023, doi:10.3390/children10040738_

Round 1

Reviewer 1 Report

This paper is describing the protocol for a RCT of a school-based web-based(?) lifestyle intervention to increase adherence to the Mediterranean diet and to 24-movement behaviors in children aged 6-13y old.

Comments:

1.            The objective of this manuscript is to describe the protocol for a RCT and this has to be clearly stated as it is currently missing from both the abstract but also in line 129 before the specific study objectives are presented. Please refer to this example: https://bmjopen.bmj.com/content/12/2/e054594.long

2.            Abstract line 56: “The primary objective…the Saludable project on 24- movement behaviors…”; the adherence word is missing before 24-h movement behaviors?

3.            The title could also mention the study population?  E.g. schoolchildren?

4.            Lines 56, 129-130: “Archena Infancia Saludable project” describes the whole RCT and not just the intervention part. The aim is to determine the 6-month effects of a lifestyle program or intervention (or how you can call it more accurately) on adherence…This is a minor comment that the authors may consider.

5.            Line 155:  Will first phase be concluded within 2 months?  It sounds a highly ambitious aim if protocols not already prepared. Since this is not an application for funding this is of less importance, yet still this timeline seems a bit unrealistic.

6.            Lines 186-18:  The authors report that multiple schools have been considered but line 185 says one school will only serve for recruitment. So all recruitment will be restricted to one school?  How representative of the Spanish schoolchildren population will this sample be? Will ethnicities, socioeconomic status etc. will be compared to national trends?  In general, will it be generalizable?

7.            How can this intervention be blinded to nutritionists, PA, physicians, nurses etc. since they have to deliver the program?

8.            Line 192:  how does risk of bias is decreased by randomizing after the baseline data collection? Maybe this will cause selection bias by including participants with certain characteristics in specific groups?

9.            The Intervention is not described adequately in section 2.3. A major objective of this paper is to present the intervention that will be implemented but here we are provided with very little information on the material that will be used to deliver it. For example, will it all be exclusively delivered online or face to face sessions will also be held? Only the parents will go through the intervention material? The kids will not be provided with any material? Will you somehow check whether parents go through the contents and send reminders if they do not? Will you measure compliance to the intervention material itself (eg. How much of the material they read or go through?)

10.        Will both parents/legal guardian follow those guidelines? If parents do not themselves change their own habits, how will their kids do since the parents seem to be those that need to deliver the information from the interventional program to their kids?

11.        Why will the interventional educational material be provided only to parents/guardians than both parents and children?

12.        The authors mention in lines 210-213 that the material of the intervention will be created. The timeline seems even less feasible considering that this has not been created yet. 2022-2023 is the year that this study will be implemented (so in reality it has already started and is midway?) but the material is not yet prepared?

13.        The drop-out rate of 10% seems low. Most studies expect 20% drop-out rate. Do you have evidence to make this argument stronger, for a 10% drop-out rate to be expected in this population?

14.        Will primary outcomes testing adherence to Med Diet and 24-h behavior be self-reported and also proxy-reported?  Since this is a major outcome, it might be better for both parents and children to report on adherence? Can you specify which are self-reported and which proxy-reported?

15.        It would make for a more robust design if anthropometrics (especially BMI), but also adherence to Med Diet and 24-h movement behaviors was also assessed in parents/guardians since the family environment might be a strong predictor to adherence?

16.        Sleep disorders outcome filled by participants or parents/guardians too? Same for physical fitness, so since you mention it is self-reported you mean by the participant themselves? In general, please specify were proxy-reported too.

17.        There is no external funding for this study. How will costs for this project be met if there is no funding available?

18.        Line 89: perhaps “studied” instead of “analyzed”?

19.        Perhaps MedDiet or another acronym would be better than MD? As a reader I keep reading MD as Medical Doctor, especially in the US!

20.        112 line: Any adherence rates to med diet that can be reported?

21.        126 line does not read well, especially where it says “including both components in combination with one another”.  Similarly, lines 515-516. You mean interaction effects when both are implemented?

Reviewer 2 Report

This protocol deals with a nutrition-related issue that is on the WHO agenda of important health topics deserving of evidence-based interventions. In this study protocol, the authors provide a comprehensive and integrated project named “Archena Infancia Saludable” which includes both interventions and process evaluation. The aims of this study are very interesting and helpful, and the results may facilitate implementing widescale health promotion strategies among schoolchildren.

I have a few remarks:

Vocabulary such as ”Notwithstanding” (line 99), although correct may not be as easily understood by non-native readers. Maybe choose a more common but stil formal alternative.

Lines 166-167: I suggest avoiding unnecessarily complex syntax ”The selection of this age group is justified by the fact that childhood” = because (”This project will be targeted at children aged 6-13 years because childhood is a critical period...”).

Line 185: Why just one school? How do the authors plan to account for the specificity of that particular school to ensure that the results are representative of the wider study population? 

Reviewer 3 Report

Thank you for the opportunity to review this manuscript.

I found huge merit in the rationale developed over the text, but some adjustments are needed before I can recommend it for publication in the Journal.

Please, see some specific comments below:

In the phrase: "Unfortunately, despite being an evidence-based healthy dietary pattern, a systematic review has pointed out the clear trend of decreasing adherence to MD in Mediterranean countries, especially among children [16,17].", there is a lacking in pointing out the possible reasons for that or at least some hypothesis to explain the reasons...

Is there a registration number for this RCT? Please, mention that and the platform used.

Please include a more adequate nomenclature for those aged 13 years. I do not consider them as children but as adolescents. 
